# Human Fallopian Tube-Derived Organoids with TP53 and RAD51D Mutations Recapitulate an Early Stage High-Grade Serous Ovarian Cancer Phenotype In Vitro

**DOI:** 10.3390/ijms25020886

**Published:** 2024-01-10

**Authors:** Yilin Dai, Jing Xu, Xiaofeng Gong, Jinsong Wei, Yi Gao, Ranran Chai, Chong Lu, Bing Zhao, Yu Kang

**Affiliations:** 1Obstetrics and Gynecology Hospital of Fudan University, Shanghai 200011, China; 2Shanghai Key Laboratory of Female Reproductive Endocrine Related Diseases, Shanghai 200011, China; 3State Key Laboratory of Genetic Engineering, School of Life Sciences, Human Phenome Institute, Fudan University, Shanghai 200438, China

**Keywords:** fallopian tubes, high-grade serous ovarian cancer (HGSOC), organoids, RAD51D

## Abstract

RAD51D mutations have been implicated in the transformation of normal fallopian tube epithelial (FTE) cells into high-grade serous ovarian cancer (HGSOC), one of the most prevalent and aggressive gynecologic malignancies. Currently, no suitable model exists to elucidate the role of RAD51D in disease initiation and progression. Here, we established organoids from primary human FTE and introduced TP53 as well as RAD51D knockdown to enable the exploration of their mutational impact on FTE lesion generation. We observed that TP53 deletion rescued the adverse effects of RAD51D deletion on the proliferation, stemness, senescence, and apoptosis of FTE organoids. RAD51D deletion impaired the homologous recombination (HR) function and induced G2/M phase arrest, whereas concurrent TP53 deletion mitigated G0/G1 phase arrest and boosted DNA replication when combined with RAD51D mutation. The co-deletion of TP53 and RAD51D downregulated cilia assembly, development, and motility, but upregulated multiple HGSOC-associated pathways, including the IL-17 signaling pathway. IL-17A treatment significantly improved cell viability. TP53 and RAD51D co-deleted organoids exhibited heightened sensitivity to platinum, poly-ADP ribose polymerase inhibitors (PARPi), and cell cycle-related medication. In summary, our research highlighted the use of FTE organoids with RAD51D mutations as an invaluable in vitro platform for the early detection of carcinogenesis, mechanistic exploration, and drug screening.

## 1. Introduction

high-grade serous ovarian cancer (HGSOC) is a frequently encountered and highly aggressive gynecological malignancy, contributing to a significant proportion (70–80%) of fatalities related to ovarian cancer [1]. A predominant proportion of HGSOC cases are diagnosed during advanced phases, marked by abdominal cavity metastasis, thereby confining the comprehensive assessment of early stage tumor progression [2,3].

HGSOC primarily originates in the distal fallopian tube rather than ovarian surface epithelial (OSE) cells. This notion is supported by the presence of shared mutations of the tumor suppressor, TP53, among serous tubal intraepithelial carcinoma (STIC), HGSOC, and TP53 signature lesions [4,5,6], and HGSOC as well as fallopian tube epithelial (FTE) cells show similar transcriptional patterns and proteomic profiles [7,8,9,10,11].

TP53 mutations are present in approximately 96% of HGSOC cases [3]. Aberrations in the homologous recombination (HR) pathway can play a pivotal role in the advancement of HGSOC [3,12,13]. Pathogenic mutations in RAD51D, an HR component and a member of the RAD51 paralog, have been identified as crucial risk factors for HGSOC, with some sites having greater impact among East Asian patients [14,15,16,17,18]; however, factors such as murine embryonic lethality have limited the development of appropriate mutational in vitro or in vivo models [15]. Previous in vivo studies have shown that deleting TP53 can extend the embryonic lifespan of RAD51D-deficient mice by 6 days and prevent the cell lethal phenotype. This effect is accompanied by pronounced chromosomal instability and heightened centrosome fragmentation [19].

Organoids, cultivated through cell sorting and subsequent growth within three-dimensional (3D) systems, provide an advanced in vitro model. Compared with traditional two-dimensional (2D) cultures and RAD51D-deficient xenograft (PDX) models, organoids exhibit enhanced genetic stability over extended periods, a closer preservation of source tissue or organ characteristics, and a faithful replication of malignant progression [20]. Especially when exploring the early stages of tumorigenesis, organoids provide a valuable platform for the precise observation and comparative analysis of the diverse functions exerted by various oncogenic factors during tumor progression.

Mouse or human-derived FTE organoids harboring mutations (e.g., TP53 ± BRCA1) have been created to study the impact of these specific mutations on early stage HGSOC [21,22,23,24,25,26]; however, little data are available on organoids with RAD51D mutations.

Thus, a novel human FTE organoid model with the knockdown of TP53 ± RAD51D was generated during the current work and used to study the in vitro impact of RAD51D.

Upon incorporating TP53 deletion, the organoids with TP53 + RAD51D knockdown (TP53 + RAD51D-KD) exhibited enhanced proliferation and organoid formation efficiency compared with the RAD51D knockdown (RAD51D-KD) group. Simultaneously, there was a reduction in ciliary differentiation, cell senescence, and apoptosis, replicating an early-stage HGSOC phenotype in vitro. In comparison to the RAD51D-KD group, TP53 + RAD51D-KD organoids showed similar levels of double-strand breaks (DSBs). Nevertheless, TP53 + RAD51D-KD organoids exhibited a significant decrease in cell senescence and apoptosis, and protective mechanisms such as G1-related cell cycle arrest mediated by TP53 were inactivated, elucidating the mechanism of genomic instability and FTE lesion generation. The significant sensitivity of TP53 + RAD51D-KD organoids to Nutlin3a (MDM2 inhibitor), carboplatin, PARPi, and cell cycle-related medication underscored the profound role of FTE organoids in drug sensitivity screening.

## 2. Results

### 2.1. Establishment of Human Fallopian Tube-Derived Organoids and the Stable Knockdown of TP53 and RAD51D

FTE organoids exhibited similar spheroid structures under bright-field microscopy (Figure 1A) and maintained morphological stability throughout over ten passages and cryopreservation. Morphological characteristics and the expression of FTE markers were assessed through the application of hematoxylin and eosin (H&E) staining (Figure 1B) as well as immunofluorescence staining (Figure 1C). PAX8, a marker of fallopian tube secretory cells, and ace-α-tubulin, a marker of ciliated epithelium, were both expressed by all organoids.

In order to investigate the crucial roles of TP53 and RAD51D in HGSOC, we conducted gene knockdown experiments through shRNA targeting these genes in FTE organoids. Western blotting confirmed the absence of the expression of TP53 and RAD51D in organoid cells (Figure 1E–H).

### 2.2. Effects of TP53 and RAD51D Knockdown on Organoid Proliferation and Ciliated Cell Differentiation

By enzymatically digesting FTE organoids into single cells and assessing cystic organoid diameter (Figure 2A–C), along with ATP-generation assays (Figure 2D) and Ki67-positive cell analysis (Figure 2E,F), we examined the proliferation and viability of organoids with specific genotypes. RAD51D-KD organoids demonstrated reduced cell proliferation compared with the wild-type (WT) group, while TP53 knockdown (TP53-KD) cells exhibited increased proliferation. TP53 deletion reversed the adverse effects of RAD51D deletion on the proliferation rate in TP53 + RAD51D-KD organoids.

Organoids are generated from induced pluripotent stem cells (iPSCs) or organ-restricted adult stem cells (ASCs) [20,27]. A greater number of organoid cysts signifies enhanced organoid formation efficiency, indicating the heightened self-renewal capacity of the corresponding ASCs. Currently, evaluating organoid formation efficiency post-digestion into single cells has become a common method for assessing organoid stemness [21,28,29]. The heightened number of organoid cysts in the TP53 ± RAD51D-KD organoids, compared with the single KD of RAD51D, suggests a potential enhancement in their ability to self-renew and sustain stemness in vitro (Figure 2A,H).

STIC and HGSOC are acknowledged to lack the expression of ciliated cell markers while being positive for PAX8 [21,23,30]. In cases of tubal intraepithelial carcinomas and in individuals diagnosed with invasive serous cancer, undifferentiated stem-like FTE cells have been situated at the basal layer, lacking the typical markers of ciliated cells [31]. Based on the ace-α-tubulin level, an epithelial marker, RAD51D-KD modestly decreased the ciliary proportion in FTE organoids compared with the WT group. However, the TP53 + RAD51D-KD group exhibited a more pronounced reduction in ciliary differentiation compared with the RAD51D-KD group. This effect is likely attributed to the TP53 deletion, given that the levels of ace-α-tubulin positivity did not show a significant difference between TP53 + RAD51D-KD and TP53-KD organoids (Figure 2G,I). Moreover, gene set enrichment analysis (GSEA) based on RNA-seq data from WT and TP53 ± RAD51D-KD organoids revealed a notable reduction in the “motile cilium assembly” gene set within TP53 ± RAD51D-KD organoids (Figure 2J).

### 2.3. Impact of TP53 ± RAD51D Knockdown on Homologous Recombination, Cell Senescence, and Apoptosis in Human Fallopian Tube Organoids

Chromosomal instability is generally acknowledged to characterize HGSOC [12,32]. DSBs were generated in DNA by using mitomycin C (MMC), and γH2AX levels were measured to gauge the activity of HR. The number of cells positive for γH2AX increased in both RAD51D-KD and TP53 + RAD51D-KD organoids, and there was no noticeable difference between these two groups (Figure 3A,B). Thus, RAD51D-KD appears to adversely affect HR efficiency.

The DNA damage response (DDR) is known for its capacity to trigger replication checkpoints, leading to a pause in the cell cycle. This phenomenon has the potential to initiate cellular senescence [33]. Cellular senescence was assessed in TP53 ± RAD51D-KD FTE organoids using β-galactosidase staining (Figure 3C) and p16 protein expression (Figure 3D,E). The observed upregulation of these two markers indicated an elevation in the level of cellular aging. Thus, RAD51D-KD organoids exhibited elevated levels of senescence, which were reduced in the TP53 + RAD51D-KD organoids.

The impaired DSB repair following dysfunctional HR causes generalized genome instability, including chromosomal loss, rearrangements, and apoptosis [34]. Levels of cleaved caspase3 and caspase3/7 Glo assays were measured as metrics of apoptosis (Figure 3F,G). There was evidence of increased apoptosis in RAD51D-KD organoids, which was reversed in TP53 + RAD51D-KD organoids.

### 2.4. Cell Cycle in TP53 ± RAD51D Knockdown Organoids

DSBs in mammalian cells elicit checkpoint responses that halt cell cycle progression [35]. Flow cytometry measurements indicated an increase in RAD51D-KD and TP53 + RAD51D-KD organoid cell numbers in G2/M phases with a decreased number in G0/G1 phases. Detailed flow cytometry plots are presented in Appendix A. In the light of previous work in this area, the current findings suggested that the accumulation of DSBs caused by RAD51D knockdown led to G2/M phase arrest (Figure 4A) [36,37]. A greater number of cells in the TP53 + RAD51D-KD organoids were present in the S and G2/M phases compared with those in RAD51D-KD organoids (Figure 4A).

Western blotting analysis showed a decrease in p21 after TP53 + RAD51D-KD and elevated levels of G0/G1 phase proteins (CDK2 and CCNE1) in TP53 + RAD51D-KD organoids. Levels of G2/M phase proteins (CCNB1 and CDK1) showed no difference between TP53 + RAD51D-KD and RAD51D-KD, but were decreased compared with WT organoids (Figure 4C,D). Comprehensive details regarding the primary antibody used can be found in Appendix A. TP53 KD appeared to reduce the occurrence of G0/G1 phase arrest, allowing DNA replication to occur so that cells accumulated DSB damage and genome instability. The DNA replication pathway exhibited enrichment in TP53 + RAD51D-KD organoids, as indicated by GSEA results from RNA-seq data, thus reinforcing this conclusion (Figure 4B).

### 2.5. Transcriptomic Profiles of Fallopian Tube Organoids of Varied Genotypes

We conducted RNA sequencing on FTE organoids with varying genotypes (Figure 5A). Kyoto Encyclopedia of Genes and Genomes (KEGG) and gene ontology (GO) enrichment analysis showed the enrichment of upregulated immune–related pathways, such as antigen processing and presentation, IL-17 signaling, cytokine-cytokine receptor interaction, and chemokine signaling in TP53 + RAD51D-KD organoids compared with the WT (Figure 5B). Pathways associated with inflammation, such as allograft rejection, TNF signaling, and the NF-κB pathway (Figure 5B), and with proliferation, such as DNA replication (Figure 4B), were also enriched.

Pathways shown to be downregulated in TP53 + RAD51D-KD organoids included those related to cilia assembly, development, and motility (Figure 5D). Features of cilia have previously been shown to disappear during HGSOC tumorigenesis [21,24,38], a finding which is consistent with the current results on ace-α-tubulin staining (Figure 2G,I).

### 2.6. TP53 + RAD51D Knockdown Upregulating the IL-17 Signaling Pathway

The IL-17 signaling pathway is pivotal in tumor progression [39,40]. IL-17A, which is predominantly synthesized by immune cells, particularly Th17 cells, has been linked to carcinogenesis and the self-renewal capacity of HGSOC stem-like cells [41,42,43]. IL-17C, another member of IL-17 family synthesized by epithelial cells, initiates an autocrine loop within epithelial tissues, which consequently upregulates IL-17A expression in TH17 cells and promotes cancer cell proliferation [44,45,46]. We obtained HGSOC patient tumor tissues and confirmed the elevated expression of IL-17A (Figure 6A) and IL-17C (Figure 6B) via reverse transcriptase polymerase chain reaction (qRT-PCR) compared with normal FTE tissues from patients with benign gynecological diseases. Immunohistochemistry (IHC) further validated the elevated expression of IL-17C in HGSOC tissues (Figure 6C).

To determine the influence of the IL-17 signaling pathway on FTE organoids, we administered IL-17A to the conditioned media. A concentration of 50 ng/mL of IL-17A caused a significant increase in cell viability for WT organoids, but no notable impact on TP53 + RAD51D-KD organoids (Figure 6D). The impact mechanism of IL-17 on tumor cells involves direct modulation through the regulation of inflammatory factors, chemokines, and growth factors [39,40]. We confirmed the upregulation of IL-17-signaling-pathway-related molecules in TP53 + RAD51D-KD organoids compared with the WT group by using enzyme-linked immunosorbent assays (ELISAs) (Figure 6E), qRT-PCR (Figure 6F), and RNA-seq (Appendix A). It was observed that, while shTP53 and shRAD51D alone were capable of upregulating their expression to some extent, they did not achieve the level observed in TP53 + RAD51D-KD (Figure 6F).

### 2.7. Sensitivity of TP53 ± RAD51D Knockdown Organoids to Platinum-Based PARP Inhibitors and Cell cycle-related Medication

Ovarian tumors with flawed HR mechanisms exhibit heightened vulnerability to PARPis, as these compounds prompt the formation of DSBs either directly or by inducing the stalling and subsequent breakdown of replication forks [47,48]. Mutant organoids were assessed for sensitivity to carboplatin, olaparib, niraparib, and rucaparib. TP53 + RAD51D-KD organoids were more sensitive to carboplatin and PARPis (olaparib, niraparib, and rucaparib) than WT and TP53-KD organoids (Figure 7B–E), consistent with the findings of previous clinical studies [49,50,51]. TP53-KD and TP53 + RAD51D-KD organoids were also more resistant to the MDM2 inhibitor, nutlin3a, than WT (Figure 7A).

Prior research has revealed that HGSOC cells with CCNE1 amplification display elevated CDK2 expression, and reducing or inhibiting CDK2 could lead to decreased ovarian cancer cell proliferation [52,53]. Given the elevated levels of CDK2 and CCNE1 in TP53 + RAD51D-KD organoids in contrast with the RAD51D-KD group, we examined how a CDK2 inhibitor (SNS-032) affected the organoids. TP53 + RAD51D-KD organoids exhibited increased sensitivity to SNS-032 compared with both the WT and RAD51D-KD groups (Figure 7F), demonstrating the profound potential of FTE organoids for high-throughput in vitro drug screening, and for investigating the impact of different mutations on cell cycle protein inhibitors and other potential drugs in anticancer treatment.

## 3. Discussion

Pathogenic RAD51D mutations have been implicated in the transformation of normal FTE cells into HGSOC. Despite its importance, the precise role of RAD51D in the early onset and progression of the disease remains largely unexplored owing to a lack of suitable models. Using advanced 3D culture techniques, we have successfully developed organoid models that originate from normal FTE. These cutting-edge models offer significant advantages over traditional 2D cell lines or PDX models for studying tumorigenic mutations and their impact on malignancy. By implementing genetic manipulation methods, we are able to delve deeper into understanding the molecular characteristics, drug responses, and other assessments related to TP53 and RAD51D mutations in FTE lesion generation in vitro.

HGSOC is characterized by a defective HR pathway and systemic TP53 mutations. Our study not only demonstrated that the combination of TP53 and RAD51D mutations induced an early HGSOC phenotype in vitro within the FTE, but also investigated the underlying molecular mechanisms, building on previous investigations. RAD51D knockdown in the organoids resulted in HR impairment, leading to DSB accumulation and G2/M phase arrest, whereas, in the TP53 + RAD51D group, TP53 knockdown mitigated apoptosis and reduced G0/G1 phase arrest. Previous studies have indicated that, when DSBs occur, checkpoint responses are activated, halting cell cycle progression and resulting in arrest; however, the thresholds of DNA damage necessary for activating DNA damage checkpoints and imposing cell cycle arrest in mammalian cells are high. Therefore, DNA lesions caused by replication stress are frequently transported into the subsequent G1 phase [35,54,55]. As a consequence, they have a greater reliance on G1 checkpoints to induce cell cycle arrest when subjected to minimal DNA damage [35,52,54,55,56]. In breast cancer cells lacking BRCA2, the TP53 pathway is activated by spontaneous DSBs, causing G1 phase arrest and cellular senescence [35]. TP53 is known to regulate G1 phases, limiting cell division and proliferation when DNA damage occurs, whereas TP53 mutations diminish G0/G1 phase arrest and apoptosis, causing replication fork damage, genomic instability, and DSB accumulation [53,57,58]. This is particularly important for the FTE, as previous research has suggested that the FTE has a weaker response and repair capacity to replication stress than the OSE [59]. In addition, the FTE is more susceptible to irritants such as follicular fluid [60].

Organoids have great potential for high-throughput in vitro drug screening, facilitating the identification of drugs with specific efficacy against malignant tumors in future research. As HR-deficient ovarian tumors exhibit heightened sensitivity to PARP inhibitors, these agents prompt the generation of DSBs either directly or through the stalling and subsequent collapse of replication forks [47,49,50,51]. TP53 + RAD51D-KD organoids in our study were thus more sensitive to carboplatin and PARPis than WT organoids. FTE organoids in our research exhibited different expression levels of cell cycle-related proteins, including p21, CCNE1, and CDK2, depending on the mutations they carried. While our observations indicated heightened sensitivity of TP53 + RAD51D-KD organoids to the CDK2 inhibitor (SNS-032) compared with both the WT and RAD51D-KD groups, which might be attributed to the elevated expression of CCNE1 and CDK2 in TP53 + RAD51D-KD organoids, it is essential to acknowledge a limitation of our study—the restricted scope of drug sensitivity assessments. The effectiveness of CDK4/6 inhibitors in clinical research highlights the significance of targeting cell cycle proteins in anticancer treatments [61,62]. HGSOC cells exhibiting CCNE1 amplification display elevated CDK2 expression, and reducing or inhibiting CDK2 leads to decreased ovarian cancer cell proliferation [63,64]. The CDK7 inhibitor also demonstrated significant cytotoxicity against ovarian tumors [65]. Such genetic susceptibilities for cancer therapy may be identified more readily by the generation of mutation-specific organoids.

Previous research and our study have shown that using shRNA, Ad-Cre, or CRISPR/Cas9 techniques to induce various disruptions in human or mouse FTE organoids, such as TP53 ± BRCA1, can lead to diverse effects on organoid characteristics, including cilia development and cell polarity [21,24,26]. The discrepancies in these results may be attributed to the residual protein expression associated with shRNA compared with the complete KO achieved with alternative techniques, and also to the functional disparities between different HR-related proteins. For instance, RAD51D contributes to the assembly of RAD51 nucleoprotein filaments, whereas BRCA1 has diverse functions in DSB processing [15]. BRCA1 plays a crucial role in regulating the cell cycle and DNA repair pathways by interacting with MRN and CtIP complexes to facilitate the processing of DSBs; it prevents 53BP1 from entering DSB sites and acts as a scaffold for the more than 100 proteins involved in these processes [66,67,68,69]. HR-associated genes contribute differently to the absolute risk of HGSOC. RAD51C presents a risk range of 10–15% compared with 10–20% for RAD51D and 39–58% for BRCA1 [16,70]. BRCA1 mutations are predominantly associated with estrogen receptor-negative breast tumors, but BRCA2 mutations are not linked to any specific pathological subtype of breast cancer [71]. Recent research demonstrates that individuals harboring BRCA2 mutations have a higher risk of developing male breast cancer, prostate cancer, and pancreatic cancer than those with BRCA1 mutations [72]. These mutations are potential candidates for future studies utilizing FTE organoids to evaluate their individual impacts on HGSOC.

The IL-17 signaling pathway plays an essential role in tumor progression, exerting its influence on tumor cells through multiple avenues; for instance, IL-17A, which is predominantly synthesized by immune cells, particularly Th17 cells, has been linked to HGSOC carcinogenesis by upregulating MTA1 and enhancing the self-renewal capacity of CD133-positive HGSOC stem-like cells [41,42,43]. IL-17C, another cytokine belonging to the IL-17 family, is predominantly synthesized by epithelia rather than hematopoietic cells [45]. It triggers an autocrine loop within epithelial tissues and stimulates the production of IL-17A [44,45]. IL-17C plays a crucial role in facilitating the infiltration of tumor-associated neutrophils and promoting the proliferation of lung cancer cells [46]. Our study not only validated the increased expression of IL-17A and IL-17C in HGSOC compared with FTE tissues, but also revealed that the administration of IL-17A to the conditioned media significantly increased cell viability in WT organoids, while having no significant effect on TP53 + RAD51D-KD organoids. Regarding the underlying mechanism for the IL-17 signaling pathway to manipulate tumor progression, previous studies demonstrate that it directly impacts these cells by overseeing the control of inflammatory factors, spurring tumor cell proliferation, fostering the epithelial-to-mesenchymal cell transformation (EMT), upregulating MMP, attracting inflammatory cells to tumor sites, and thwarting the processes of autophagy and cell death [39,40,73,74]. IL-17A upregulated CXCL1 secretion in breast cancer cells, resulting in the enhanced migration, invasion, and increased expression of pAKT and pNF-κB [75]. Chemokines associated with the IL-17 signaling pathway, including CXCL1, CXCL2, CXCL5, CXCL8, and CCL20, have been demonstrated to promote the proliferation, migration, and metastasis of HGSOC cells [76,77,78,79,80,81,82,83]. We also confirmed the upregulation of CXCL1 in TP53 + RAD51D-KD by using qRT-PCR and ELISAs in our study. Given that various IL-17-related inhibitors and IL-17R-related inhibitors have entered clinical trial testing in recent years, a limitation of our study is the lack of investigation of the effects of these medications on TP53 ± RAD51D organoids [74]; however, our organoid model serves as a faithful physiological in vitro platform for future drug screening studies.

## 4. Materials and Methods

### 4.1. Epithelial Progenitor Isolation from Fallopian Tube Samples

This study was conducted in adherence to the principles outlined in the Declaration of Helsinki. Ethical approval was obtained from the Ethical Institutional Review Board of the Obstetrics and Gynecology Hospital of Fudan University (approval number: 2021-174, 27 September 2021). All participants provided informed consent for the utilization of their tissue samples in scientific research.

Anatomically normal fallopian tubes from 16 patients who underwent standard surgical procedures (salpingectomy) for benign gynecologic diseases were used within 2–3 h after surgical removal (patients’ clinical data are shown in Appendix A).

Human HGSOC samples were obtained from 10 patients undergoing primary debulking procedures or recurrent metastasis biopsy (patients’ clinical data are shown in Appendix A).

The origin of the cell lines and organoids used was verified via short tandem repeat (STR) profiling.

### 4.2. Organoid Culture

Organoids were successfully cultivated using established methods [21,22,23,24,25,26,84]. The oviducts underwent a digestion process with a solution containing 0.125 mg/mL collagenase sourced from clostridium histolyticum (Sigma, St. Louis, MO, USA), 0.125 mg/mL Dispase II (Life Technologies, Gaithersburg, MD, USA), and 10 µM of Y27632 (Tocris, Bristol, UK) at 37 °C for a duration of 1 h with shaking. This was followed by mechanical shearing and a 5 min centrifugation at 450× *g*. A suspension of cell-Matrigel was applied to a pre-warmed culture plate and allowed to solidify for 15 min at 37 °C before introducing the specific culture medium (Appendix A).

### 4.3. Cloning of shRNAs and Virus Production

shRNA sequences designed to target specific segments of TP53 and RAD51D mRNA were inserted into the lentiviral vector pLKO, which included blasticidin or puromycin resistance genes for mammalian selection (Genomeditech Co., Ltd., Shanghai, China).

The following target sequences were used to knockdown TP53 and RAD51D:

TP53: 5′-CACCATCCACTACAACTACAT-3′.

RAD51D: 5′-CCTGTGCTGTTGTTTGGGAAA-3′.

Lentiviral particles were generated by human embryonic kidney cells (HEK-293T). After 48 h of transient transfection, lentiviral supernatants were harvested, filtered, and subsequently concentrated through centrifugation.

All cell lines used were purchased from the American Type Culture Collection (ATCC, Manassas, VA, USA).

### 4.4. Western Blotting

Cell pellets were resuspended in lysis solution (Beyotime, Shanghai, China) supplemented with protease inhibitors (Biotool, Houston, TX, USA). Samples were separated on 10% SDS-PAGE gel and then transferred to 0.22 μm PVDF membranes (Millipore, Billerica, MA, USA). The membranes were then immersed in a 5% milk solution for 60 min and exposed to primary antibodies for 16 h at 4 °C before adding HRP-conjugated secondary antibodies for 1 h at room temperature. After the membranes were washed, chemiluminescence reagents (Yeason Shanghai, China) were added and detection was performed using the Chemi Doc Imaging System (Bio-Rad, Hercules, CA, USA). Details of the antibodies used are shown in Appendix A.

### 4.5. Organoid Growth Assay

Organoids were detached from Matrigel and digested with TrypLE (Thermo Fisher, Waltham, MA, USA). Following trypsinization, cells were passed through a 70 μm cell strainer to obtain a single-cell suspension. One thousand cells/well were plated into 96-well plates. Organoids were captured using a high-throughput cell analyzer (Countstar Castor X1, Alit Life Science, Shanghai, China), and their diameter as well as formation efficiency were determined via a circle Hough transform analysis in ImageJ.

ATP assays were executed by utilizing a CellTiter-Glo-3D cell viability test kit (Promega, Madison, WI, USA). To evaluate the impact of the IL-17 pathway on FTE organoids, cell viability was assessed following a 72 h treatment with recombinant human IL-17A (PeproTech, Rocky Hill, NJ, USA) at concentrations of 0, 10, and 50 ng/mL. Chemiluminescence was quantified by using a microplate reader (PerkinElmer, Waltham, MA, USA).

### 4.6. Immunofluorescent and IHC

For the 4 µm FFPE sections, antigen retrieval was carried out by using a steamer and citrate buffer (0.1 mol/L, pH of 6.0). The sections were exposed to primary antibodies (details presented in Appendix A) at 4 °C overnight and probed with secondary antibodies (Beyotime, Shanghai, China) before being subjected to a DAPI-containing mounting medium (Servicebio, Wuhan, China). Images were captured with an FV3000 confocal laser scanning microscope (Olympus GmbH, Hamburg, Germany) and processed with Photoshop [CZ1] Software Version 2020 (Adobe, San Jose, CA, USA).

Sections were exposed to primary antibodies (details presented in Appendix A) overnight at 4 °C. For negative controls, sections were left without primary antibody incubation. Goat anti-rabbit secondary antibody conjugated with horseradish peroxidase (Vector Laboratories, Burlingame, CA, USA) was utilized. The chromogenic reaction was executed with DAB (Vector Laboratories, Burlingame, CA, USA).

### 4.7. Flow Cytometry and Cell Cycle Analysis

The dissociation of organoids into a single-cell suspension was accomplished through trypsinization. The cells were then stained with a cell cycle staining kit (CCS012, LIANKE BIO, Hangzhou, China). The analyses of these samples were conducted by using a BD FACSCalibur system (BD Bioscience, San Jose, CA, USA) and ModFit LT software version 5.0.9.

### 4.8. Cell Senescence and β-galactosidase

Organoids were stained by using a β-galactosidase staining kit (G1073, Servicebio, Wuhan, China). The fixation of organoids was carried out at room temperature by using a β-galactosidase staining fixative for a duration of 15 min. A β-galactosidase staining solution was introduced, and the organoids were subjected to a 6 h incubation at 37 °C. bright-field observation was performed.

### 4.9. Apoptosis Caspase-Glo3/7 3D Assay

Oviduct organoids were enzymatically dissociated into single cells, seeded onto a black 96-well plate, cultured for 14 days, and assayed by using a Caspase-Glo3/7-3D kit (G8091, Promega, Madison, WI, USA). A Caspase-Glo3/7-3D buffer was added to the provided lyophilized substrate and vortexed vigorously. The reagent was introduced to each well and incubated for 30 min at room temperature in a light-protected environment. The amount of chemiluminescence was measured by using a microplate reader (PerkinElmer, Waltham, MA, USA).

### 4.10. Drug Sensitivity Assays

Organoids were seeded in 96-well plates on day 0 at an initial seeding density of 1000 cells per well. Rucaparib (S1098, Selleck, Houston, TX, USA), niraparib (S2741, Selleck, Houston, TX, USA), olaparib (S1060, Selleck, Houston, TX, USA), carboplatin (S1215, Selleck, Houston, TX, USA), nutlin3a (HY-10029, MedChem Express, Monmouth Junction, NJ, USA), and SNS-032 (S1145, Selleck, Houston, TX, USA) were added on day 7 at the indicated concentrations. The culture medium was refreshed, and a new drug dose was introduced every 3.5 days. The cell viability was assessed on day 14 by using a CCK8 assay kit (K1018, APExBIO, Houston, TX, USA) with incubation at 37 °C for 4 h. Measurements were performed using a microplate reader (PerkinElmer, Waltham, MA, USA), and the results were normalized to DMSO controls.

### 4.11. RNA Extraction and qRT-PCR

RNA extraction was performed by using an RNAprep Pure Micro Kit (DP420, Tiangen, Beijing, China). RNA was reverse transcribed by using a Goscript Reverse Transcription System (Promega, Madison, WI, USA). Each PCR was performed in a volume of 15 μL by using an SYBR Green Master Mix (Selleck, Houston, TX, USA) in triplicate on a CFX384 Touch System (Bio-Rad, Hercules, CA, USA). GAPDH served as the internal control gene in all qRT-PCR analyses. Details of the primers used are shown in Appendix A.

### 4.12. RNA Sequencing and Analysis

Total RNA was extracted from organoids using an Ovation RNA-seq System V2 Kit (NuGEN) to create libraries for deep sequencing.

The extraction and library preparation processes were carried out in a biosafety level 3 facility and in compliance with strict regulations. The libraries were sequenced on an Illumina NovaSeq 6000 platform. Clean reads were aligned to the human reference genome (GRCh38) with the use of HISAT2 (version 2.1.0) after undergoing quality assessments. The alignments were then processed using StringTie (version 2.2.1) to assemble each sample’s transcripts and compare the splicing results to known transcripts. HTSeq (version 0.9.1) compared the Read Count values for each gene as the original expression. FPKM was utilized to standardize the gene expression values. DESeq (version 1.38.3) was used for differentially expressed gene (DEG) analysis. Genes were identified as DEGs if they met the following criteria: (1) absolute log2 (fold change) ≥ 1 and (2) *p*-value < 0.05. We utilized pheatmap (version 1.0.12) to generate visualizations and the hierarchical clustering of log2-transformed FPKM data.

To investigate the functional roles of the DEGs, we conducted GO enrichment analyses and explored the KEGG pathways by using ClusterProfiler. *p* < 0.05 was used as the cutoff criterion [85]. Additionally, we employed GSEA to further investigate the biological characteristics of the two risk groups. Significance was determined by *p*-values < 0.05.

### 4.13. ELISA

To quantify the CXCL1 released by FTE organoids in culture supernatants, we conducted an ELISA. Initially, the organoids were dissociated into single cells by using enzymatic techniques, and equal cell counts were cultured in 24-well plates. The conditioned medium was collected after a 72 h incubation period on day 14 and cleared of cellular debris via centrifugation. The CXCL1 levels were assessed by using a CXCL1/GRO alpha Quantikine ELISA Kit (R&D systems, Minneapolis, MN, USA).

### 4.14. Statistical Analysis

Experiments were performed at least in triplicate on organoids from 3 different patients. The conclusions were drawn based on the data obtained through GraphPad Prism v. 8.0 software (GraphPad, San Diego, CA, USA). The results were presented as a mean value with an accompanying standard error of the mean (SEM).

*p*-values were calculated using a one-way analysis of variance (ANOVA) followed by a Tukey test, unless otherwise specified. Paired *t*-tests were utilized to compare the mean band intensity values of the Western blotting. The cell cycle distribution was assessed by a two-way ANOVA followed by the Tukey test. A Mann-Whitney *t*-test was employed to compare the mRNA levels of IL-17A, IL-17C, CXCL1, CXCL2, and CXCL6, whereas Welch’s *t*-test was used for CXCL3, CXCL5, CXCL8, and CCL20. An unpaired two-tailed Student’s *t*-test was applied to evaluate the CXCL1 released from FTE organoids in culture supernatants. Half-maximal inhibitory concentration (IC_50_) values were derived from nonlinear regression curves (dose-response curves).

Statistical significance was set at ns (not significant), *p* > 0.05; * *p* < 0.05; ** *p* < 0.01; *** *p* < 0.001; and **** *p* < 0.0001.

## 5. Conclusions

In conclusion, a novel human FTE organoid model, modified using lentiviral shRNA to knockdown TP53 ± RAD51D, was generated, and the impacts of the mutations were analyzed in vitro. TP53 deletion rescued the adverse effects of RAD51D deletion on the proliferation, stemness, senescence, apoptosis, and cell cycle progression of FTE organoids. Our research highlighted the utility of FTE organoids with RAD51D mutations as an invaluable in vitro platform for the detection of early carcinogenesis, mechanistic exploration, and drug screening.

## Figures and Tables

**Figure 1 ijms-25-00886-f001:**
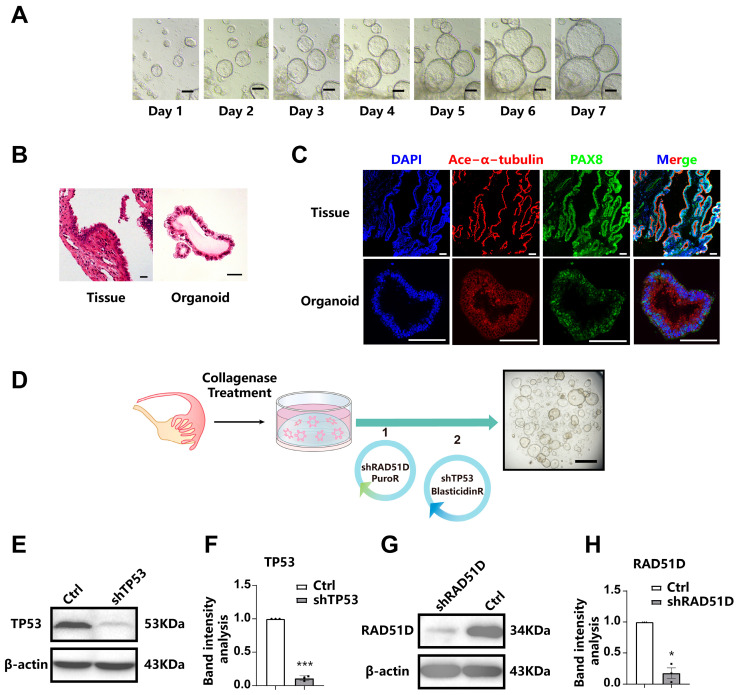
Establishment of human fallopian Tube-Derived organoids and the stable knockdown of TP53 and RAD51D. (**A**) bright-field microscopy showing the spheroid morphology of FTE organoids. Scale bar: 100 μm. (**B**) H&E staining of FTE organoids and oviduct tissues. Scale bar: 100 μm. (**C**) Immunofluorescence staining of FTE organoids and fallopian tube tissues, showing PAX8 (green), ace-α-tubulin (red), and DAPI (blue). Scale bar: 100 μm. (**D**) A diagram outlining the approach for creating mutated organoids utilizing shRNA. Scale bars: 1 mm. (**E**–**H**) Western blot analysis and associated statistical evaluation of the band intensity values of TP53. (**E**,**F**) and RAD51D (**G**,**H**). *, *p* < 0.05; and ***, *p* < 0.001.

**Figure 2 ijms-25-00886-f002:**
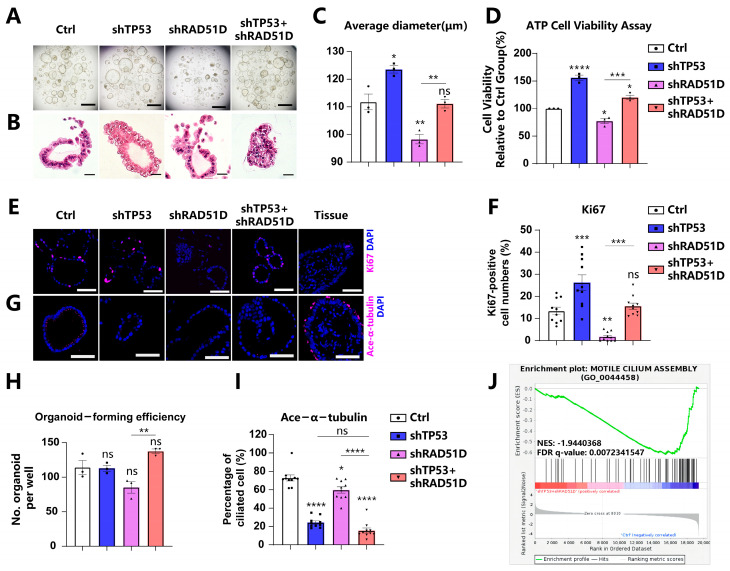
Effects of TP53 and RAD51D knockdown on organoid proliferation and ciliated cell differentiation. (**A**,**B**) Bright field (**A**) and H&E staining (**B**) images depicting TP53 ± RAD51D-KD FTE organoids. Scale bars: 1 mm (**A**) and 100 μm (**B**). (**C**) Organoid diameter after TP53 ± RAD51D-KD. (**D**) ATP generation of TP53 ± RAD51D-KD organoids. (**E**,**F**) Immunofluorescence staining of Ki67 (**E**) and ratios of positive cells (**F**) in TP53 ± RAD51D-KD FTE organoids. Scale bar: 50 μm. (**G**,**I**) Immunofluorescence staining of ace-α-tubulin (**G**) and ratios of positive cells (**I**) in TP53 ± RAD51D-KD FTE organoids. Scale bar: 50 μm. (**H**) Evaluation of cyst formation of TP53 ± RAD51D-KD FTE organoids. (**J**) GSEA of WT versus TP53 ± RAD51D-KD organoids for motile cilium assembly. NES—normalized enrichment score, FDR—false discovery rate. ns—not significant, *p* > 0.05; *, *p* < 0.05; **, *p* < 0.01; ***, *p* < 0.001; and ****, *p* < 0.0001.

**Figure 3 ijms-25-00886-f003:**
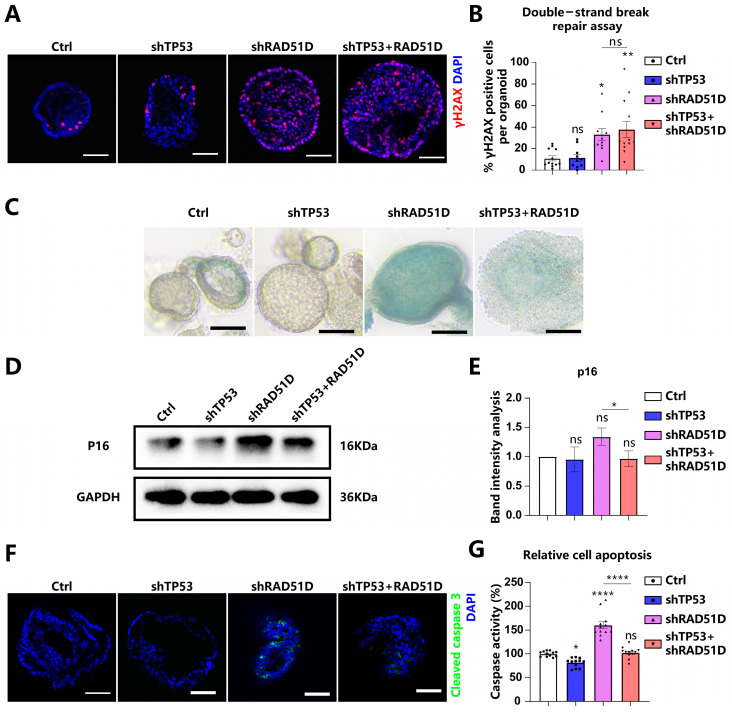
The knockdown of TP53 ± RAD51D exerted a notable influence on the function of HR, cell senescence, and apoptosis in FTE organoids. (**A**,**B**) TP53 ± RAD51D-KD organoids were cultured overnight with MMC. γH2AX was detected via immunofluorescence analysis (**A**) and the percentage of γH2AX-positive cells was analyzed (**B**). Scale bar: 50 μm. (**C**) bright-field images of β-galactosidase staining of TP53 ± RAD51D-KD FTE organoids. Scale bar: 100 μm. (**D**,**E**) Western blot analysis (**D**) and associated statistical evaluation of the band intensity values (**E**) of p16 expression in TP53 ± RAD51D-KD FTE organoids (*n* = 3). (**F**) Immunofluorescent measurement of cleaved caspase3 in TP53 ± RAD51D-KD FTE organoids. Scale bar: 50 μm. (**G**) Quantification of apoptosis in TP53 ± RAD51D-KD FTE organoids using a caspase3/7 Glo assay. ns—not significant, *p* > 0.05; *, *p* < 0.05; **, *p* < 0.01; and ****, *p* < 0.0001.

**Figure 4 ijms-25-00886-f004:**
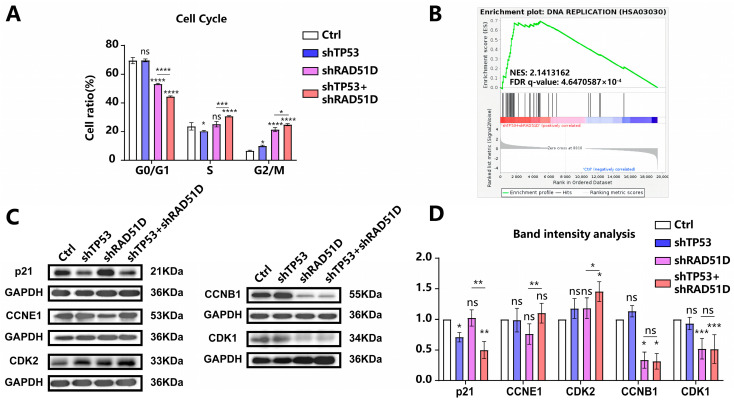
Cell cycle in TP53 ± RAD51D-KD organoids. (**A**) The cell cycle distribution in TP53 ± RAD51D-KD organoids (*n* = 3) was analyzed using flow cytometry. (**B**) GSEA demonstrated the enrichment of the DNA replication pathway in TP53 + RAD51D-KD organoids. NES—normalized enrichment score, FDR—false discovery rate. (**C**,**D**) Western blotting analyses (**C**) and associated statistical evaluation of the band intensity values (*n* = 3) (**D**) were conducted to assess the protein expression related to cell cycle checkpoint regulation. ns—not significant, *p* > 0.05; *, *p* < 0.05; **, *p* < 0.01; ***, *p* < 0.001; and ****, *p* < 0.0001.

**Figure 5 ijms-25-00886-f005:**
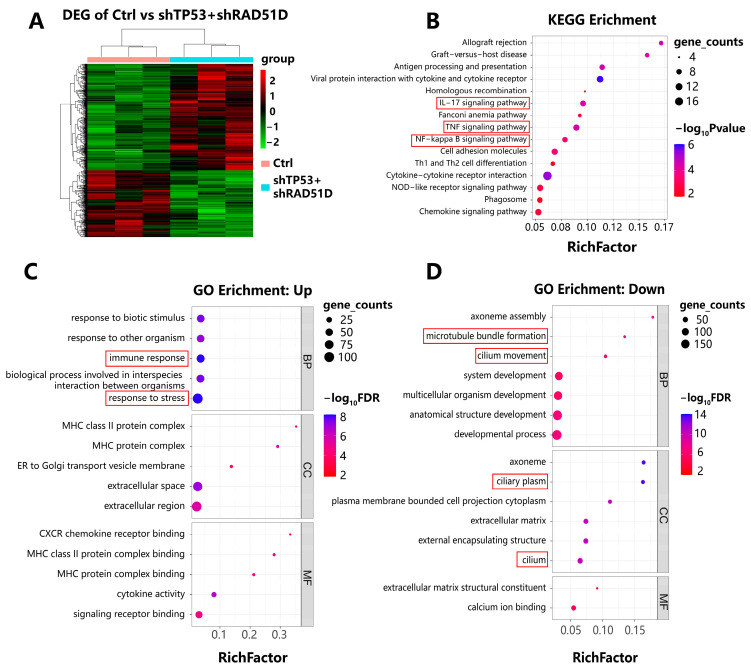
Transcriptomic profiles of FTE organoids of varied genotypes. (**A**) A two-way clustering heatmap illustrating differentially expressed genes in both WT and TP53 + RAD51D-KD organoids (*n* = 3 per group). (**B**) KEGG analyses were performed among the indicated groups, ranked by the rich factor. Shading depicts gene counts, whereas color denotes the *p*-value within each category. (**C**,**D**) GO enrichment analyses between the indicated groups, ranked by rich factor. Shading represented gene counts, whereas color indicated the FDR-adjusted *p* value within each category. (**C**) Upregulated pathways in TP53 + RAD51D-KD organoids. (**D**) Downregulated pathways in TP53 + RAD51D-KD organoids. FDR—false discovery rate.

**Figure 6 ijms-25-00886-f006:**
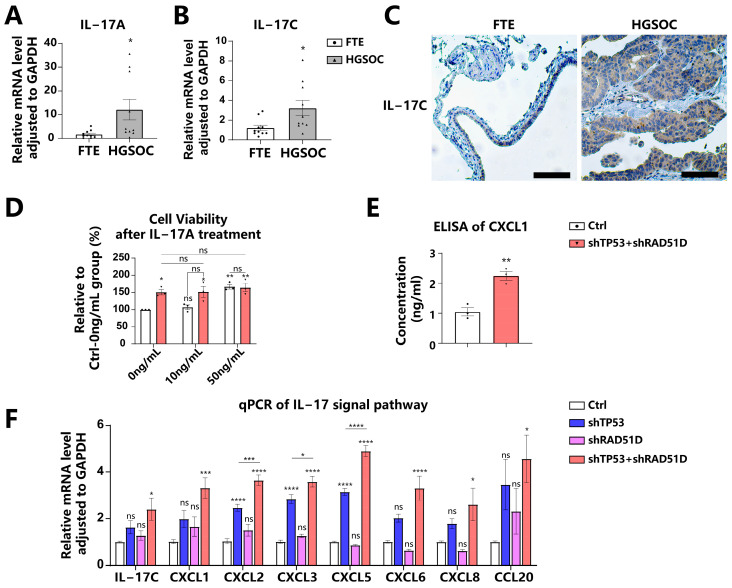
TP53 + RAD51D knockdown upregulated IL-17 signaling pathway. (**A**,**B**) We utilized qRT-PCR to evaluate the expression of IL-17A (**A**) and IL-17C (**B**) in FTE and HGSOC tissues. (**C**) IHC staining of IL-17C in FTE and HGSOC tissues. Scale bar: 100 μm. (**D**) Cell viability of WT and TP53 + RAD51D-KD organoids after IL-17A treatment for 72 h (0, 10, and 50 ng/mL). (**E**) ELISA of CXCL1 in 72 h conditioned media supernatants from the indicated organoids. (**F**) qRT-PCR was utilized to assess the mRNA level of IL-17 signaling pathway-related genes in TP53 ± RAD51D-KD organoids (*n* = 6). ns—not significant, *p* > 0.05; *, *p* < 0.05; **, *p* < 0.01; ***, *p* < 0.001; and ****, *p* < 0.0001.

**Figure 7 ijms-25-00886-f007:**
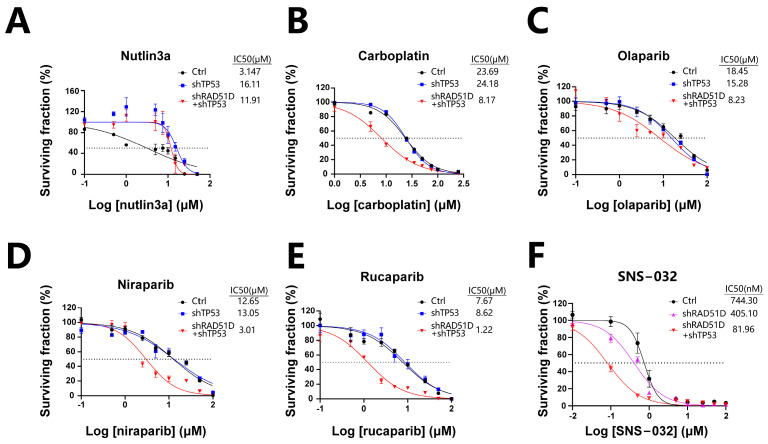
TP53 ± RAD51D knockdown affected sensitivity to platinum, PARPis, and cell cycle-related therapies. (**A**) TP53-KD and TP53 + RAD51D-KD organoids were more resistant to nutlin3a than the WT. (**B–E**) TP53 + RAD51D-KD organoids were more sensitive to carboplatin (**B**) and PARPis (olaparib, niraparib, and rucaparib) (**C–E**) than the WT and TP53-KD organoids. (**F**) TP53 + RAD51D-KD organoids were more sensitive to the CDK2 inhibitor (SNS-032) than the WT and RAD51D-KD groups. Data were presented as the mean ± standard error of the mean (SEM).

## Data Availability

All data and materials are available from the corresponding author upon request.

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
