# Peer review of "Human Fallopian Tube-Derived Organoids with TP53 and RAD51D Mutations Recapitulate an Early Stage High-Grade Serous Ovarian Cancer Phenotype In Vitro"

_ijms, 2024, doi:10.3390/ijms25020886_

Round 1
Reviewer 1 Report
Comments and Suggestions for Authors
1. It is not clear what clinical problem does the manuscript address. Is it progression of ovarian cancer? Considering that only 0.6% of ovarian cancer cases carry RAD51D mutations the clinical relevance is uncertain. If the objective of the manuscript is to see the effect of an acquired somatic p53 mutation on the background of pre-existing RAD51D germline mutation in high-risk women – the emphasis should be made on comparing cells carrying a single RAD51D mutation with double mutants, although comparison with single p53KD could also be useful outside the scope of this study.
2. The malignant potential of RAD51D and double KD should be demonstrated in animal model, or referred to if already reported.
3. It is not described how the study revealed that FTE organoids harboring TP53 and RAD51D mutations replicate an early‐stage HGSOC phenotype in vitro, specifically what parameters were used for proof.
4. Statement on lines 65-68 needs to be clarified.
5. The interpretation of results provided in Figure 2C-F should be combined as all apply to cell proliferation/viability and all 3 experiments are concordant.
6. In Figure 2A double-KD organoid structure seems to be different from both single KDs. Is this a consistent event? If so – it needs to be acknowledged.
7. It is not clear how the conclusion is made that the higher number of organoid cysts in the TP53±RAD51D‐KD cells compared to the single RAD51D-KD suggests a potential enhancement in their ability to self‐renew and sustain stemness in vitro.
8. Immuno-fluorescence staining of aaTUBB for shRAD51D in Fig.2G looks strange, positive cells seemingly do not coincide with inner sphere diameter. The results in Figures 2G-I - should first describe the effect of RAD51D KD on ace-a tubulin expression, then addition of p53 KD, and finally effect of p53 alone used as an explanation of the combined effect.
9. P.172 – reference to Table 4 needs to be inserted after describing the effects on cell cycle progression phase.
10. Figure 4A should be also presented as a flow cytometry plot.
11. Sentence on pp.172-174 “in the light…” – why the conclusion is made ‘in light of … findings’? The experimental evidence presented by authors indicates that RAD51D KD led to G2/M phase arrest. References seem to be irrelevant.
12. P.173 – typo ‘leaded’ has to be corrected to ‘led’.
13. Sentence on pp.177-179 ‘Western blotting analysis…” should be corrected for grammar. Something like “Western blotting analysis showed a TP53‐dependent decrease in p21 and elevated levels of G0/G1 phase proteins (CDK2 and CCNE1) in TP53+RAD51D‐KD organoids”. Where is the evidence that decrease in p21 is TP53-dependent?
14. Are the data from 2J and 4B derived from RNAseq? This should be mentioned in the text describing both figures.
15. Figure 6. Is expression of cytokines involved in IL17 signaling pathway is also elevated in RNAseq data? Why the results for single KD are not shown?
16. Figure 7. A-D do not show RAD51D, but shows TP53. All four types of organoids should be presented for each drug. If no difference is noted – it should be ‘data not shown’ in text.
Comments on the Quality of English LanguageThere are several typos and other orthographical and grammatical errors that should be corrected.
Author Response
Dear reviewer,
Thank you for reviewing our manuscript and providing these constructive comments. We have gone through them one by one and made revisions in the manuscript accordingly. Please see the attachment. Following is our response (in blue font) to each of your comment. We have also specified the revision we made in the response. Please review and let us know your thoughts.
Best regards!
Yours sincerely,
Yu Kang
Reviewer #1:
Major Comments:
- It is not clear what clinical problem does the manuscript address. Is it progression of ovarian cancer? Considering that only 0.6% of ovarian cancer cases carry RAD51D mutations the clinical relevance is uncertain. If the objective of the manuscript is to see the effect of an acquired somatic p53 mutation on the background of pre-existing RAD51D germline mutation in high-risk women – the emphasis should be made on comparing cells carrying a single RAD51D mutation with double mutants, although comparison with single p53KD could also be useful outside the scope of this study.
Response: Thank you for your insightful suggestion.
Our primary aim is to investigate the impact of an acquired somatic TP53 deletion on the background of a pre-existing RAD51D germline mutation in high-risk women, with a specific focus on phenotypic manifestations associated with the progression of ovarian cancer.
Responding to your suggestion, additional experiments comparing single deletions with double mutants have been included in the newly submitted revised version (e.g., question 15). In the original manuscript, we also conducted a comparative analysis between RAD51D-KD and TP53+RAD51D-KD organoids, evaluating parameters such as proliferation, organoid formation efficiency, homologous recombination (HR), senescence, apoptosis and cell cycle progression.
Regarding the pathogenicity of RAD51D deleterious mutations in HGSOC, a clinical study involving 7216 families with a history of HGSOC revealed significant associations between pathogenic variants in RAD51D and HGSOC (J Natl Cancer Inst. 2020;112(12):1242-1250. doi:10.1093/jnci/djaa030). RAD51D exhibited a relative risk (RR) of 7.60 (95% CI = 5.61 to 10.30; P = 5 × 10-39), with an estimated cumulative risk of developing HGSOC by age 80 years at 13% (95% CI = 7-23%) for individuals with pathogenic RAD51D variants. Therefore, we deem it crucial to focus on phenotypes associated with the progression of HGSOC.
- The malignant potential of RAD51D and double KD should be demonstrated in animal model, or referred to if already reported.
Response: Thank you for the comment. The previously reported animal models have been referenced in the introduction of the resubmitted revised version.
- It is not described how the study revealed that FTE organoids harboring TP53 and RAD51D mutations replicate an early‐stage HGSOC phenotype in vitro, specifically what parameters were used for proof.
Response: Thank you very much for your insightful advice.
Firstly, TP53+RAD51D-KD organoids displayed heightened cell proliferation, increased organoid formation efficiency, and decreased cilia development—common features observed in tubal intraepithelial carcinomas and HGSOC. In contemporary studies on malignant transformation, evaluating FTE cilia proportion and organoid formation efficiency post-digestion into single cells has become a standard practice (e.g., EBioMedicine. 2020;56:102800. doi:10.1016/j.ebiom.2020.102800, Figure 3; Cancer Discov. 2021;11(2):362-383. doi:10.1158/2159-8290.CD-20-0455, Figure S2).
Additionally, prior studies suggest that combining TP53 gene mutation with HRD-related gene mutation significantly leads to accumulated chromosomal instability, a characteristic molecular feature of HGSOC. RAD51D-KD and TP53+RAD51D-KD organoids exhibited comparable levels of DSBs indicated by γH2AX. However, there was a significant reduction in cell senescence and apoptosis in TP53+RAD51D-KD organoids, and protective mechanisms such as G1-related cell cycle arrest mediated by TP53 were inactivated. This elucidates the mechanism underlying genomic instability and FTE lesion generation. The co-deletion of TP53 and RAD51D also upregulated multiple HGSOC-associated pathways, including the IL-17 signalling pathway.
In conclusion, despite potential limitations, we believe that in this study, TP53+RAD51D-KD replicates an early-stage HGSOC phenotype in vitro.
- Statement on lines 65-68 needs to be clarified.
Response: Thank you for the comment. In the updated manuscript, the conclusions previously outlined in lines 65-68 have been thoroughly elaborated.
- The interpretation of results provided in Figure 2C-F should be combined as all apply to cell proliferation/viability and all 3 experiments are concordant.
Response: Thank you for the comment. We have restructured the results of Figure 2C-F, demonstrating that both cystic organoid diameter, ATP-generation assays and Ki67-positive cell analysis were used to detect the proliferation/viability of organoids.
- In Figure 2A double-KD organoid structure seems to be different from both single KDs. Is this a consistent event? If so – it needs to be acknowledged.
Response: Thank you for the comment. Based on our observation, the major disparities between double-KD organoids and both WT or single KD organoids were the organoid formation efficiency and organoid diameter, as quantified in Figure 2C and Figure 2H. However, TP53+RAD51D-KD organoids maintained a cystic structure and did not undergo profound structural changes, such as transitioning into a solid structure. Hence, a detailed discussion of this aspect was refrained.
- It is not clear how the conclusion is made that the higher number of organoid cysts in the TP53±RAD51D‐KD cells compared to the single RAD51D-KD suggests a potential enhancement in their ability to self‐renew and sustain stemness in vitro.
Response: Thank you for your constructive advice.
Organoids are derived from induced pluripotent stem cells (iPSCs) or organ-restricted adult stem cells (ASCs) through cell sorting and spatially restricted lineage commitment. A greater number of organoid cysts signifies enhanced organoid formation efficiency, indicating the heightened self-renewal capacity of the corresponding ASCs. Currently, in the investigation of malignant transformation and other in vitro phenotypes of organoids, evaluating organoid formation efficiency post-digestion into single cells has become a common method for assessing organoid stemness (e.g. Nat Commun. 2019;10(1):1194. Published 2019 Mar 18. doi:10.1038/s41467-019-09144-7, Figure 3). We have incorporated this background information into the corresponding section of the revised manuscript.
In addition, we also assessed the ciliary differentiation and stemness of specific organoids through the detection of ace-α-tubulin and RNA-seq enrichment. This evaluation is based on the observation that undifferentiated stem-like FTE cells, lacking typical markers of ciliated cells, are situated at the basal layer in patients with tubal intraepithelial carcinomas or HGSOC (Stem Cells. 2012;30(11):2487-2497. doi:10.1002/stem.1207). This is also a conventional method routinely employed in previous studies for analyzing the stemness of FTE organoids (e.g., Nat Commun. 2019;10(1):5367. Published 2019 Nov 26. doi:10.1038/s41467-019-13116-2, Figure 2).
- Immuno-fluorescence staining of aaTUBB for shRAD51D in Fig.2G looks strange, positive cells seemingly do not coincide with inner sphere diameter. The results in Figures 2G-I - should first describe the effect of RAD51D KD on ace-a tubulin expression, then addition of p53 KD, and finally effect of p53 alone used as an explanation of the combined effect.
Response: Thank you for your suggestion. The discrepancy in alignment between positive cells and the inner sphere diameter in Figure 2G may be attributed to slice layer. To address this, we replaced the figure with a standard immunofluorescence image, ensuring no impact on our ace-α-tubulin quantification results at the same time. We have also reorganized the ace-α-tubulin results for RAD51D-KD, TP53+RAD51D-KD and TP53-KD organoids in accordance with your recommendations in the revised version.
- P.172 – reference to Table 4 needs to be inserted after describing the effects on cell cycle progression phase.
Response: Thank you for your suggestion. In the revised manuscript, we incorporated a reference to Table S4 following the discussion of effects on cell cycle progression phases. Additionally, we specified that comprehensive details regarding the primary antibody used can be found in Supplementary Table 4 within the Western blotting section of the methodology module.
- Figure 4A should be also presented as a flow cytometry plot.
Response: Thank you for your valuable comment. In the newly submitted modified version, we present flow cytometry plots, which was originally used as raw materials, to readers as Supplementary Figure 1.
- Sentence on pp.172-174 “in the light…” – why the conclusion is made ‘in light of … findings’? The experimental evidence presented by authors indicates that RAD51D KD led to G2/M phase arrest. References seem to be irrelevant.
Response: Thank you very much for your valuable advice. After meticulous re-evaluation, we have substituted the citations with more suitable and precise references. Prior studies indicated that deleting BRCA1 or RAD51C results in G2/M phase arrest due to the accumulation of unrepaired DNA damage (Nat Struct Mol Biol. 2010;17(6):688-695. doi:10.1038/ SMB.1831; J Cell Sci. 2013;126(Pt 1):348-359. doi:10.1242/jcs.114595).
- P.173 – typo ‘leaded’ has to be corrected to ‘led’.
Response: Thank you for your comment. In the revised version, we have replaced ' leaded ' with 'led'.
- Sentence on pp.177-179 ‘Western blotting analysis…” should be corrected for grammar. Something like “Western blotting analysis showed a TP53‐dependent decrease in p21 and elevated levels of G0/G1 phase proteins (CDK2 and CCNE1) in TP53+RAD51D‐KD organoids”. Where is the evidence that decrease in p21 is TP53-dependent?
Response: Thank you very much for your valuable advice. Our original intention was to convey that shTP53 induced the downregulation of p21 expression in TP53-KD and TP53+RAD51D-KD organoids. To address potential confusion, we have omitted "TP53-dependent" in the latest revised version.
- Are the data from 2J and 4B derived from RNAseq? This should be mentioned in the text describing both figures.
Response: Thank you for your valuable suggestion. In the resubmitted manuscript, we have specified in the relevant text that figure 2J and figure 4B originate from the GSEA enrichment analysis of RNA-seq data.
- Figure 6. Is expression of cytokines involved in IL17 signaling pathway is also elevated in RNAseq data? Why the results for single KD are not shown?
Response: Thank you for your valuable suggestion.
Apart from the increase observed in qRT-PCR, the cytokines involved in the IL-17 signaling pathway also displayed upregulation in RNA-seq data. We annotated these differentially expressed cytokines on the Ctrl vs TP53+RAD51D-KD DEG volcano plot and incorporated them into the recently submitted manuscript as Supplementary Figure 2.
In the recently submitted revised manuscript, we confirmed the expression of these cytokines individually using qRT-PCR in single KD groups. It was observed that, while shTP53 and shRAD51D alone were capable of upregulating their expression to some extent, they did not achieve the level observed in TP53+RAD51D-KD organoids (Figure 6F).
- Figure 7. A-D do not show RAD51D, but shows TP53. All four types of organoids should be presented for each drug. If no difference is noted – it should be ‘data not shown’ in text.
Response: Thank you for your constructive advice. The notation 'data not shown' has been applied at the relevant location in the revised manuscript. No distinction is observed between RAD51D-KD and TP53+RAD51D-KD.
- Comments on the Quality of English Language: There are several typos and other orthographical and grammatical errors that should be corrected.
Response: Thank you for your valuable suggestion. We have employed professional Language Editing Services provided by MDPI to enhance the English language expression of this article.

Reviewer 2 Report
Comments and Suggestions for Authors
The work was carried out at a good methodological level to assess the consequences of RAD51D-restricted KD or in combination with TP53 KD in in vitro HGSOC model. I would like to raise some minor issues.
1. HGSOC can be associated with RAD51D and RAD51C mutations. Why did Authors focused only on RAD51C? What is the prevalence in RAD51C mutations in HGSOC patients? The Abstract lacks specific arguments to support that.
2. Did Authors analyze the prognostic value of RAD51D to support the above-mentioned role of this gene (e.g., using TCGA data)?
3. It would be beneficial to enlarge the size of bar charts since now they are hard to follow. Also, the resolution in particular charts at figure 5 is too low.
4. Individual data are plotted only on some of the charts, but not all of them (Figure 3B). Please unify that or provide the number of samples (biological samples or replicates) in figure description.
5. The discussion lacks information on the possible reasons underlying different changes in organoids sensitivity to specific drugs compared to WT.
6. Lines 260, 324 – change ‘researches’ to ‘research’.
Comments on the Quality of English LanguageThe article presents quite new and interesting results, I recommend it for publication with minor modifications.
Author Response
Dear reviewer,
Thank you for reviewing our manuscript and providing these constructive comments. We have gone through them one by one and made revisions in the manuscript accordingly. Please see the attachment. Following is our response (in blue font) to each of your comment. We have also specified the revision we made in the response. Please review and let us know your thoughts.
Best regards!
Yours sincerely,
Yu Kang
Reviewer #2:
Major Comments:
The work was carried out at a good methodological level to assess the consequences of RAD51D-restricted KD or in combination with TP53 KD in in vitro HGSOC model. I would like to raise some minor issues.
- HGSOC can be associated with RAD51D and RAD51C mutations. Why did Authors focused only on RAD51D? What is the prevalence in RAD51C mutations in HGSOC patients? The Abstract lacks specific arguments to support that.
Response:
Thank you for your insightful suggestion.
A clinical study of 7216 families with a family history of HGSOC or breast cancer revealed significant associations between pathogenic variants in RAD51C and RAD51D and HGSOC (J Natl Cancer Inst. 2020;112(12):1242-1250. doi:10.1093/jnci/djaa030). RAD51C exhibited a relative risk (RR) of 7.55 (95% CI = 5.60-10.19, P = 5 × 10-40), while RAD51D displayed a similar RR of 7.60 (95% CI = 5.61 to 10.30; P = 5 × 10-39). The estimated cumulative risks of developing HGSOC by age 80 years is 11% (95% CI = 6-21%) for RAD51C and 13% (95% CI = 7-23%) for individuals with pathogenic RAD51D variants.
A meta-analysis involving 29400 ovarian cancer patients reveals a mutational frequency of 0.6260% for RAD51C and 0.4125% for RAD51D in HGSOC patients, compared to 0.1126% and 0.0597% in the control population. It is noteworthy that certain pathogenic variants, such as LRG_516t1:c.270_271dup p1:p.(Lys91fs*13), pose a notably elevated risk for the East Asian population(odds ratio [OR] = 4.89; 95%CIs:1.76–13.62; p = 0.0024) ( J Ovarian Res. 2020;13(1):50. doi:10.1186/s13048-020-00654-3).
These statistics were previously mentioned in the introduction section (lines 46-49).
Our clinical experiences with HGSOC patients harboring RAD51D mutations have sparked research interest, leading us to initiate preliminary studies on the sensitivity of these individuals to PARP inhibitors (Int J Mol Sci. 2023;24(19):14476. doi:10.3390/ijms241914476). Therefore, our emphasis is on elucidating the role of RAD51D. While acknowledging the importance of investigating and comparing the functions of RAD51C and other genes associated with the homologous recombination (HR) pathway in HGSOC, our investigation primarily revolves around RAD51D. In the discussion section of our manuscript, I anticipate the potential application of fallopian tube organoids in related research (lines 380-381). However, it is essential to clarify that our study does not specifically center on this aspect. We may address pertinent content in future researches.
- Did Authors analyze the prognostic value of RAD51D to support the above-mentioned role of this gene (e.g., using TCGA data)?
Response: Thank you very much for your advice.
Extensive clinical cohort investigations have addressed the relative and cumulative risks associated with RAD51D in HGSOC, as noted in the introduction of our manuscript (lines 46-49) and our response to question 1.
Our study primarily focused on examining the potential effects of TP53 and RAD51D mutations within fallopian tubes and their implications for carcinogenesis. Consequently, we did not explore the impact of RAD51D mutations on the prognosis of HGSOC within public databases. Upon reviewing previous researches, we observed that earlier studies revealed a correlation between the expression of RAD51B, RAD51C, RAD51D, and XRCC2 and a favorable relapse-free survival (RFS) in breast cancer (Int J Gen Med. 2022;15:4925-4936). We may examine the prognostic outcomes among HGSOC patients carrying RAD51D mutations versus those bearing mutations in other HR pathway genes in our subsequent researches.
- It would be beneficial to enlarge the size of bar charts since now they are hard to follow. Also, the resolution in particular charts at figure 5 is too low.
Response: Thank you for your constructive advice. The bar charts in the revised manuscript have been increased in size, and the resolution of Figure 5 has been enhanced.
- Individual data are plotted only on some of the charts, but not all of them (Figure 3B). Please unify that or provide the number of samples (biological samples or replicates) in figure description.
Response: Thank you for your suggestion. Individual data have been incorporated into Figure 3B in the revised manuscript, and those without individual data are provided with the number of biological replicates.
- The discussion lacks information on the possible reasons underlying different changes in organoids sensitivity to specific drugs compared to WT.
Response: Thank you for your valuable comment. In the discussion section of the resubmitted version, we briefly elucidated the factors contributing to alterations in drug sensitivity compared to the WT organoids.
- Lines 260, 324 – change ‘researches’ to ‘research’.
Response: Thank you for your valuable suggestion. In the revised version, we have replaced these 'researches' with 'research'.
